# Water Heritage in the Rural Hinterland Landscapes of the UNESCO Alto Douro Wine Region, Portugal: A Digital Humanities Approach

Gerardo Vidal Gonçalves [1,*], Dina Borges Pereira [2], Martín Gómez-Ullate [3] and André da Silva Mano [4]

1 CIDEHUS Centro Interdisciplinar de História, Culturas e Sociedades da Universidade de Évora, Largo do Marquês de Marialva 8, 7000-654 Évora, Portugal

2 AHAS Associação de História e Arqueologia de Sabrosa, 5060-422 Sabrosa, Portugal

3 Resarch Group «Musical Heritage, Culture and Education», Department of Didactics of Music, Plastic and Body Expression, Teacher Training College, Universidad de Extremadura/Escola Superior de Artes Aplicadas (ESART), 6000-243 Castelo Branco, Portugal; mgu@unex.es

4 Faculty of Geo-Information Management and Earth Observation, Universtity of Twente, 7522 Twente, The Netherlands; a.dasilvamano@utwente.nl

* Correspondence: gerardo@uevora.pt

**Abstract:** Starting from a well defined and demarcated area in space, the Corgo River, in the region of Trás-os-Montes and Alto Douro, this work seeks through Information Technologies (IT), Digital Humanities and online tools and applications (software) to provide digital images of space and water resources, anthropic elements related to water and various natural features. Based on these available resources and a set of historical cartographic documents, we provide a realistic view of the cultural and natural water landscape and how augmented reality can help archaeology interpret this landscape and its historical transformations.

**Keywords:** archaeology; wate heritage; water lanscape; photogrammetry; digital modeling; history; architecture of water

## 1. Introduction

Water management, from a historical point of view, fits naturally into the history of humanity itself. On careful reflection, for the specific case of the work presented here, rivers, streams, watercourses, estuaries, dams, wells, mechanical bridges, aqueducts, and other elements related to water aspects are essentially an extension of man's needs throughout history. On Earth, it is synonymous to life and, obviously, man has always taken advantage of this element, adjusting and conditioning his relationship with this fundamental element.

Water is an essential resource for life on Earth. More than 71% of the planet (1.4 billion $km^3$) is made up of water and, without a doubt, water management has been with man since ancient times [1].

On planet Earth, we can say that about 2.5% is freshwater and 97.5% is saltwater. In general, freshwater has very unequal percentages, where about 0.9% corresponds to the water present in soils, swamps, and frozen areas, groundwater comprises about 29.5%, rivers and lakes about 0.3%, and about 69.2% corresponds to eternal snow and ice zones [1].

Water management, from a historical point of view, fits universally into the history of humanity itself. On careful reflection, for the specific case of the work presented here, rivers, streams, watercourses, estuaries, dams, wells, mechanical bridges, aqueducts, and other elements related to water aspects are essentially an extension of man's needs throughout history.

In recent years, water-related heritage has been increasingly represented in heritage enhancement schemes—for example, through the expansion of cultural itineraries and World Heritage inscriptions—and in scholarship.

In Portugal, aqueducts have especially been an object of study, acknowledgement, and safeguard. Celia López-Bravo, José Peral López, and Eduardo Mosquera Adell enlist a series of 18 waterworks with different degrees of protection, 30% of them included in UNESCO lists [2].

In the Trás-os-Montes and Alto Douro region, in Portugal, the importance of watercourses and natural irrigation channels, streams, creeks, and others is well-recognized [3].

The region of Tras-os-Montes and Alto Douro is limited to the south by the main Iberian watercourse called the River Douro ("Durius" in Roman times), whose source occurs in a place called Serra de Urbión, in the Spanish province of Soria, flowing into the city of Porto, in the Atlantic. Despite changing material forms, the landscapes in the region epitomize successive layers of water management and human habitation, evidencing an ongoing effort to manage water for both productive (food) and preventative (flooding) purposes.

Trás-os-Montes and Alto Douro is bordered to the south by the same river, the Douro, and on the north bank, in the same region, run several watercourses, rivers, and streams. Among the most important are the Tua River, the Pinhão River, the Corgo River, and the Tâmega River. The study presented here focuses essentially on the Corgo River. The Corgo River, whose source occurs at Vila Pouca de Aguiar, at about 923 m altitude, runs from its source to the Douro river for about 53,428 m, with a maximum slope of 15.9%. On the other hand, this watercourse crosses three localities: Vila Pouca de Aguiar, more in the north; Vila Real, in the middle part of the river; and Peso da Régua, in the river's estuary, at its confluence with the Douro river.

Although it is a very relevant watercourse for understanding the settlement of the Trás-os-Montes and Alto Douro region, few historical, archaeological, or other studies on this important watercourse exist. The Corgo River, on its banks, has, throughout the history of human occupation of the territory, seen a vast, relevant, and diversified anthropic activity. In some regions of the world, the water heritage is linked to the wood industry [4]; in Portugal, it is more deeply linked to the electricity industry with several hydroelectric plants, with the oldest on Portuguese territory on the river Corgo, near the city of Vila Real, being a plant dating back to the end of the 19th century, the Biel power hydroelectric station (Figures 1–3) [5].

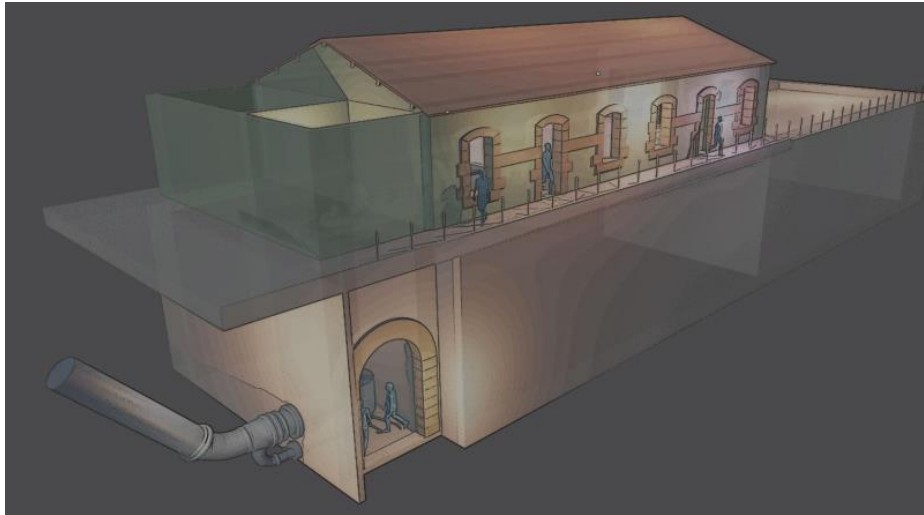

**Figure 1.** Digital modeling of the Biel Hydroelectric Power Plant, the first hydropower plant built in Portugal, still in the late 19th century, obtained from the photogrammetric survey and later modeled with Blender.

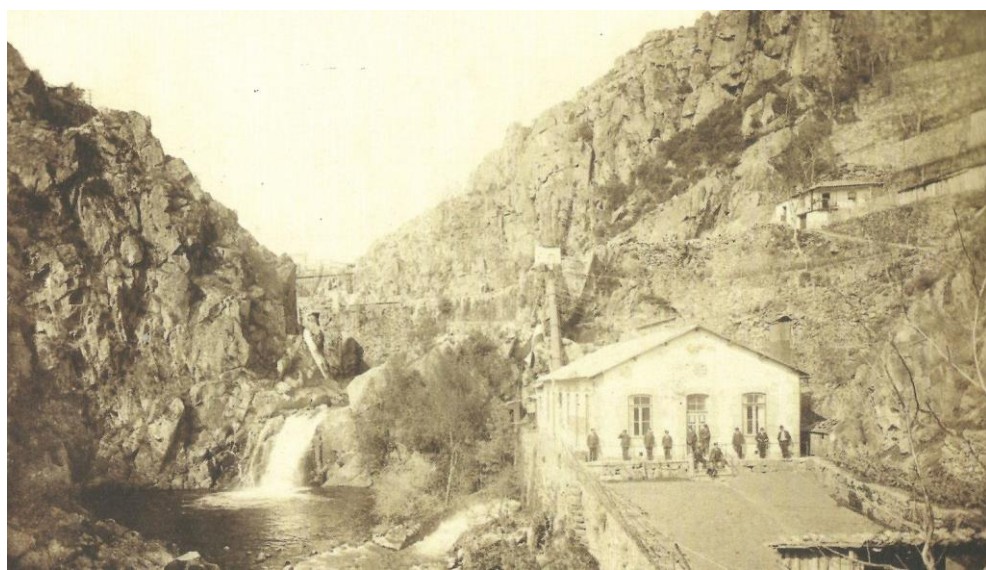

**Figure 2.** One of the first photographs of the Biel Hydroelectric Power Station, on the banks of the Corgo River, in Vila Real.

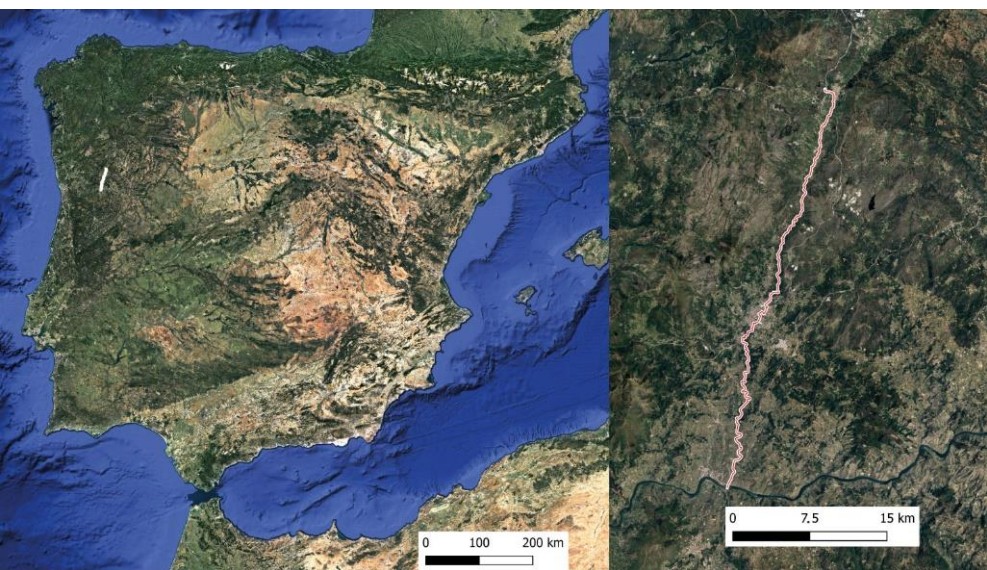

**Figure 3.** General map of the Iberian Peninsula with location of Corgo River.

The use of new information technologies, geographic information systems [6], and the digitization of artifacts, buildings [7–9], and even natural or man-made landscapes is, in essence, a fundamental resource for this study. The interpretation of the humanized landscape and the whole component related to the material and built cultural heritage is facilitated, in part, by the use of the digitization of these same natural, cultural, and historical resources.

The studies that connect the analyses or interpretations on elements of the humanized natural heritage, the architecture of water, and the material and immaterial cultural heritage for water management, in this specific case related to the water heritage or related to water and human intervention, either directly or indirectly, and the rural territory are, of course, important [10]. Understanding how man manages and controls, throughout history, one of the most important resources for his own life on Planet Earth is, in general, a herculean and never-ending task. The study of the humanized landscape and the built heritage has been primarily carried out through abstraction, using physical data. This abstraction, nowadays,

can be attenuated through the visualization of three-dimensional data obtained through the new information technologies and, above all, applied to the digital humanities.

Digital humanities has been a very interesting area of knowledge for more than 70 years [11–13]. Eminently multidisciplinary, this area of expertise has gone through several phases throughout its history [11]. Despite being a complex area, with several intersections in the domains of social sciences and humanities and computer science, the possibilities of this knowledge for interpreting data and facts in the social sciences and humanities are truly fantastic.

In the last few years, we have witnessed a revolution in the field of digital humanities [14]. The new information technologies allow us, nowadays, to complement a significant part of the treatment, analysis, research, and integrated visualization of information. From the point of view of the data collected in the domains of the various social sciences and humanities, these same data, despite eventually integrating elements of the material and physical world, when filtered, inventoried, classified, and analyzed, pass into another world, a world characterized by various abstractions and immaterialities. The transformation of an event or a physical element (historical fact, character, building, document, etc.) into something abstract and measurable mentally and analytically compromises, in part, the naturalistic, stereoscopic, and realistic vision we have of the world around us.

However, the new contributions of computing, computer engineering, multimedia, and new information technologies allow us, in part, to promote a vision, even if digital, that is a little more expressive and dynamic, of the world that surrounds us or that, naturally, surrounded us in past times.

The importance of the water factor for the preservation of biodiversity and the protection of healthy and harmonious human life on the planet, and, of course, the importance of proper water management in the context of climate change deserve special mention. Several authors, mainly in multidisciplinary areas, have addressed the relationship between water management throughout history and the current issue of climate change [15–17].

In this short article, we present some of the results of a research project focused primarily on the identification of cultural, historical, and archaeological heritage related to water, wells, dams, mills, water mills, water mills, bridges, and other mechanical devices for collecting water, around one of the important watercourses that crosses, from north to south, the region of Trás-os-Montes and Alto Douro: the Corgo River.

In this work, we explore the implementation and articulation of several techniques and methods in the digitization of historical and archaeological information to better understand the humanized landscape in the surroundings of a very expressive natural accident in the northern interior territories of Portugal. The use of new technologies, especially articulated with the digitization of physical elements, be they heritage structures or natural or man-made landscapes, through photogrammetry, has been successfully used in several works [17–19].

The articulation between the photogrammetry and multidimensional digitization of structures of archaeological, heritage, and historical relevance was complemented with the use of geographic information systems (open source) to digitize locations of elements of interest and areas of anthropic exploration in the surroundings of the Corgo River. To digitize the locations and their evolution over time, military charts and maps produced from the late 19th century (1894) were used, which had to be georeferenced using the same open-source software and aerial photography and also the aerophotogrammetric survey carried out with a drone, along the entire length of the Corgo River.

One of the main objectives of the work is, in essence, to understand, in part, the chronological and spatial evolution of anthropic investment in the construction of structures related to the capture, storage, and management of water resources in the area surrounding the Corgo River, in this first phase of 30.4 km$^2$, along the course of the river, whose length is approximately 53 km.

## 2. Materials and Methods

The region of Trás-os-Montes and Alto Douro, where minifundia predominate, is characterized by high irrigation through rivers, streams, and small watercourses, which surround and skirt the various hills, mountains, and granite outcrops characteristic of the region. It is a region with immense potential in historical and archaeological research fields. Documentary and archival information are, to date, scarce in historiographical and research processes; however, the architectural, cultural, and archaeological evidence is, indeed, vast.

It was these natural accidents, the rivers and streams, the mountains, and the fertile valleys, which, during the High Middle Ages, served as landmarks or borders for the ancient Pannonias, an administrative division of ecclesiastical character which predominated until the beginning of the Lower Middle Ages [20,21].

The preliminary study presented here seeks to complement, in part, an ancestral process in the fields of historical and archaeological research. It is important to note, however, that little has been done in the fields of integrated research, from an historical and archaeological point of view and even from the cultural environment point of view, for the region and the area under study.

For the present work, although it is a preliminary and introductory approach, several technologies were used, all of them in the fields of the so-called digital humanities, as referred and interpreted by several authors [20,22–24].

Several free computer tools and digital platforms were used, such as Google Maps, OSM, QGIS (geographic information systems), and Blender (digital modelling software). Hardware tools were also used (digital photographic equipment, digital cameras, etc.) and equipment such as ROV.s and drones, as well as a management system for the information gathered, were adapted to the preliminary needs of this study.

The vast bibliography and methodology produced in the last 10 years, concerning the implementation of new techniques of multidimensional digitalization and vectorization of physical and documentary elements, has been, naturally, of great interest [25–30].

Although, as already mentioned, this introductory work seeks to test, in the main, some online and open-source tools applied, naturally, to the digital humanities and to the understanding of part of an ancestral territory occupied by man, it was necessary to test, in the field, the symmetry of the digital data with reality. This task, still to be concluded, proved to be quite satisfactory and relevant since, in fact, the digital modelling of the territory and the vectorization of some cultural elements linked to the water factor allowed us, to a large extent, to assimilate a comprehensive and significant understanding about the territory and the human occupation of it and its management and exploitation.In fact, our field experience, especially in the several campaigns of archaeological prospection, the historical archives research and the use of new information technologies to better understand the territories and the landscape changes caused by man, was an excellent added value to integrate and understand the digital humanities and its relevance to the social sciences and humanities and other interdisciplinary knowledge areas.

## 3. Photogrammetry and Analysis of the Territory

To obtain a digital survey of the territory (the Corgo River watercourse and its surroundings) using medium altitude (500 to 700 m a.s.l.) and high-resolution images, several flight sequences were implemented to capture orthogonal aerial photography using two models of drones from the company Shenzhen DJI Sciences and Technologies Ltd. The equipment used were: (a) PHANTOM 3 4K drone, with a Sony camera and a Sony EXMOR 1/2.3″ CMOS sensor, and a FOV 94° 20 mm lens—the image capture was always performed with a resolution of 4000 × 3000 px; and (b) Drone Mavic Air 2, with a 1/2″ CMOS sensor, with 48 effective MP, and a FOV: 84° lens. Each of the obtained images was processed in photogrammetry software (Agisoft Metashape, designed by Agisoft L.L.C.) to obtain a photogrammetric alignment of the images, an optimization of the resulting alignments, and the creation of a point cloud. A digital mesh through polygons was elaborated from the point cloud and a mesh of the physical area under study was obtained. After the

elaboration of the digital mesh or mesh of the territory, a texture was created based on the chromatic registers of all the aerial photographs obtained.

The total area covered was 30,400,902.91 m$^2$ (30.4 km$^2$). The limits of this area were obtained using the Buffer tool, in the QGIS software, taking as a central reference point the vector that corresponds, approximately, to the mesial zone of the Corgo riverbed. For both sides (west and east) of this vector, an average of 300 m was considered.

A digital terrain model was also produced using images (101 photographs; resolution of 2500 × 1406 px) exported from the Google Earth Studio platform, in which flight plans were implemented at an altitude of approximately 2143 m, allowing oblique photographs to be taken in order to produce a photogrammetric model to complement the digital model obtained by taking orthogonal drone images (Figures 4–7).

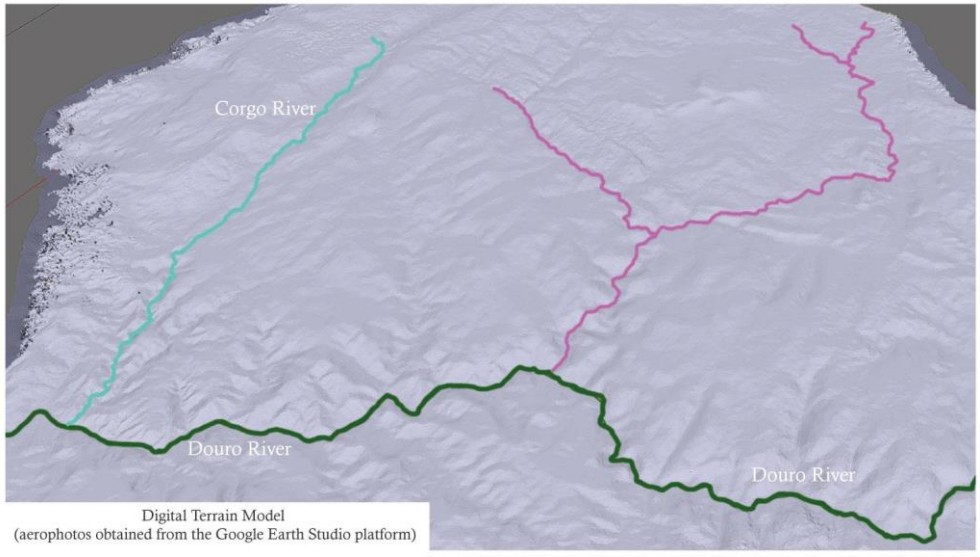

**Figure 4.** Digital model of the land of the former division of Pannonias indicating the main rivers.

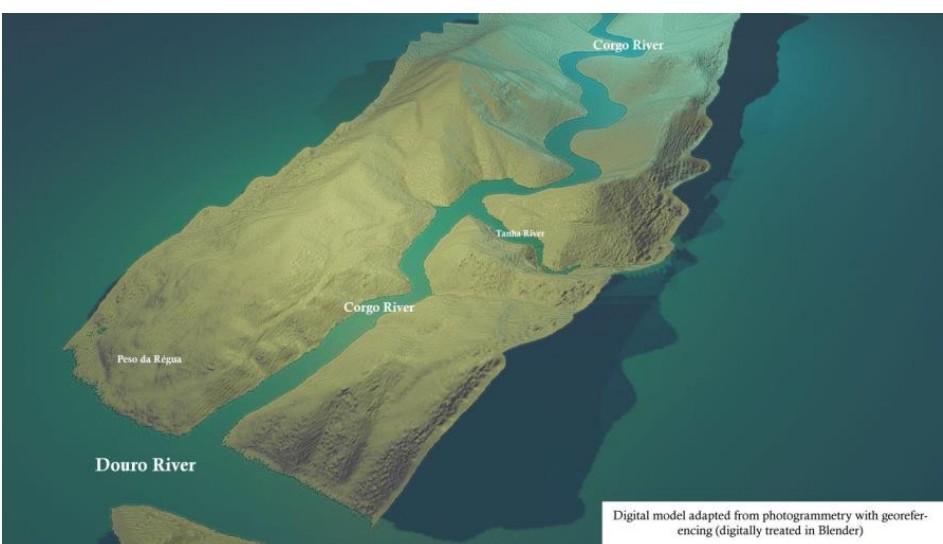

**Figure 5.** Digital model of the confluence of the Corgo River and the Douro River, obtained through photogrammetric processes and digitally treated in Blender software.

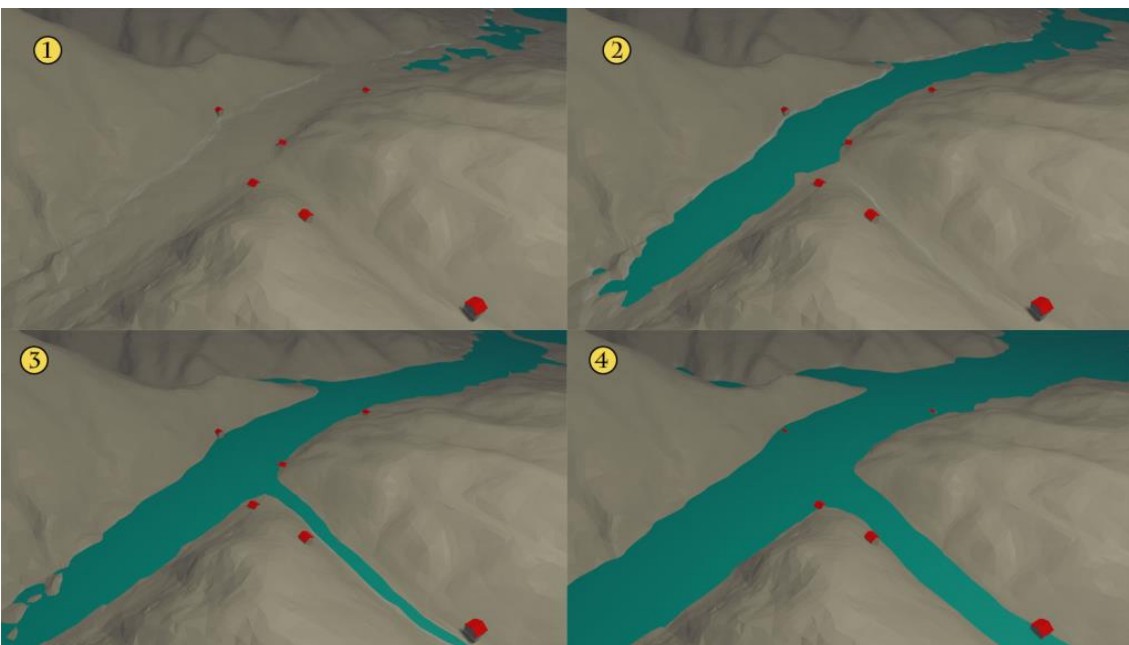

**Figure 6.** Elevation models (DEM from photogrammetry), adapted and rendered in Blender software, with the implementation of various structures such as water mills and the illustrative representation of the possible levels of rise of river waters.

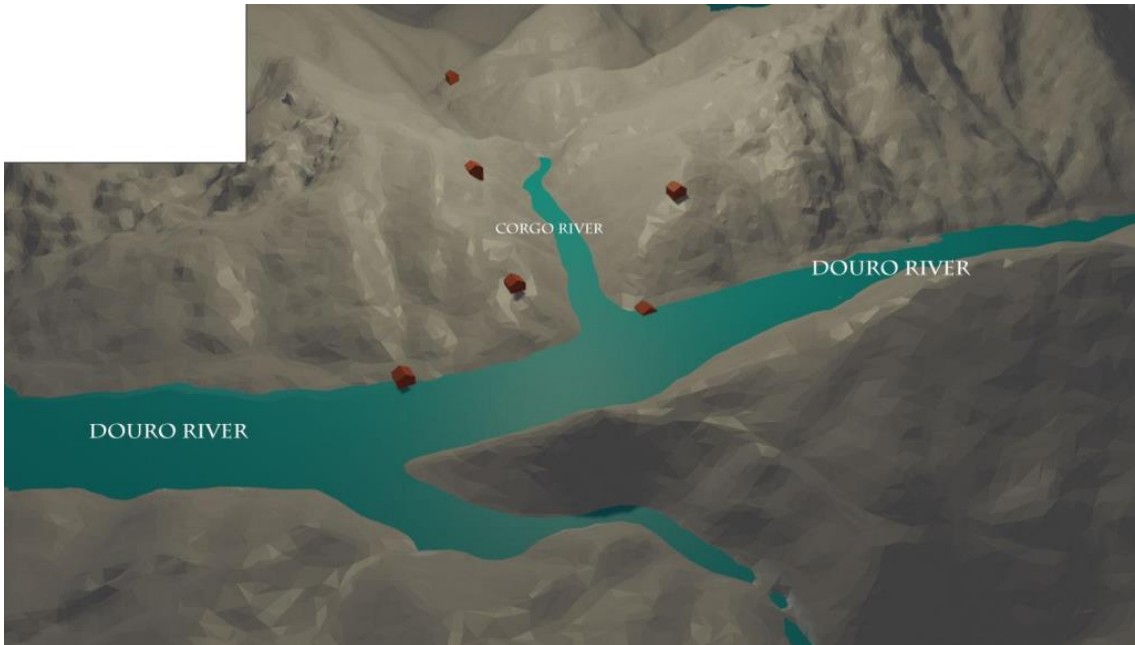

**Figure 7.** Elevation models (DEM from photogrammetry), adapted and rendered in Blender software, with the implementation of various structures such as water mills and the illustrative representation of the possible levels of rise of river waters.

Both the procedures described in the previous paragraphs and the digital results of the same operations (digital terrain models) were exported, from Agisoft Metashape, to the format "*.obj" to be subsequently treated in the open-source software Blender and in the software QGIS (Figures 5–7).

In fact, the anthropic or anthropized territory that surrounds the banks of the Corgo River, from its source, in Vila Pouca de Aguiar, to its connection with the Douro River, to

the south, is a complex territory full of evidence of ancient and modern human occupation. At the beginning of its course, in its spring, we verify, mainly through the preliminary observation of the digital terrain models and the perception, in loco, of the immense valley formed at Vila Pouca de Aguiar, which culminates, a little further south, before reaching Vila Real, a quite significant cultural reality.

Through the historical cartography and the field confirmation of some of the elements identified in the maps, more than 50 water mills, 11 water springs, seven norias, and 10 wells, or simple structures to obtain water without devices were vectorized and referenced. Around 10 bridges were also identified, some of medieval origin and a bridge of Roman origin.

The use of diverse and multidisciplinary tools and, essentially, resources in the digital humanities scope, complement this type of study. As we have already mentioned, this is a preliminary study and whose digital approach allowed, in part, to add a set of quantitative and qualitative information inexistent so far in the scientific literature for the region in question [3,5,31].

## 4. Military Cartography and Digitisation of Anthropic Features

Besides this objective, we also sought, through the creation and modeling of the physical territory and landscape, with the use of aerial photogrammetry, to obtain a generalist and less abstract vision of the area under study and its surroundings. However, historical cartography is a fundamental resource for better understanding the constructed heritage character elements.

The cartographic sources used correspond to military maps drawn up by the Portuguese Army in the last quarter of the 19th century (1895), 1930–1935, 1960–1965, and 1985–1990 (Figure 8). This cartography has several graphic elements indicating the location of man-made constructions or other environmental or natural features. Among the man-made aspects related to water, we highlight the wells, water channels, dams, fountains, bridges, etc.

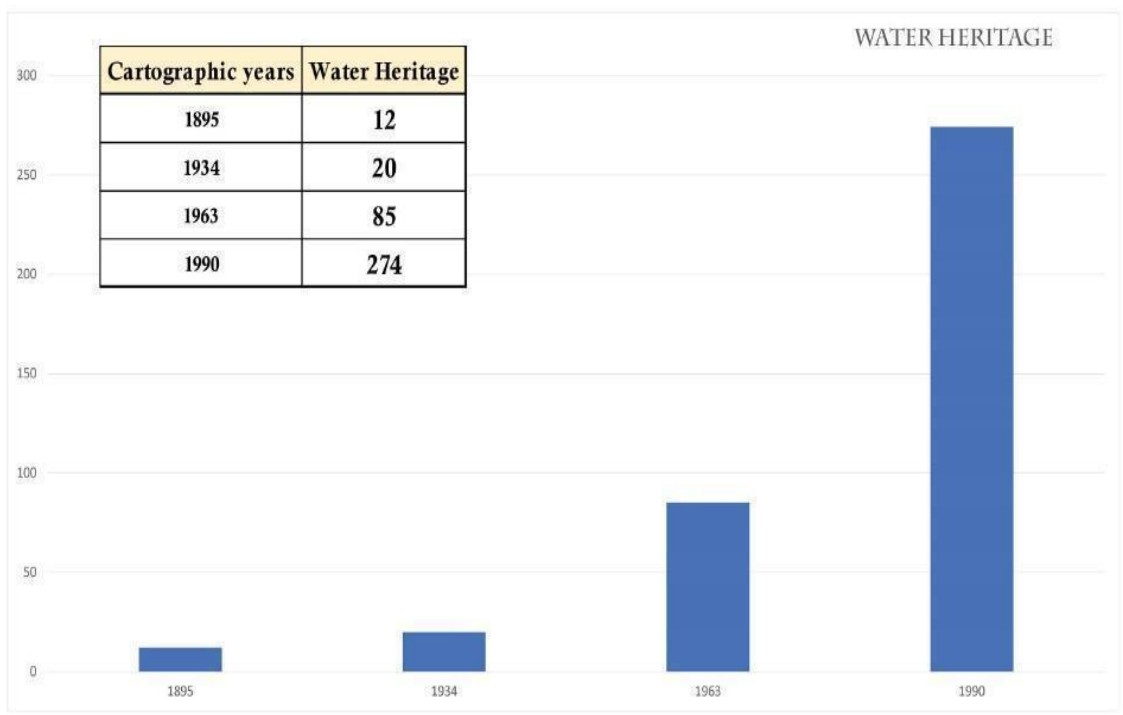

| Cartographic years | Water Heritage |
| --- | --- |
| 1895 | 12 |
| 1934 | 20 |
| 1963 | 85 |
| 1990 | 274 |

**Figure 8.** Graphic obtained from the survey, by periods and military charts, of the water heritage of the study area.

Although the cartography of the last quarter of the 20th century was already georeferenced, the older cartography, obtained from archives and scans provided by co-workers, was not georeferenced. In this sense, using the QGIS georeferencer, eight or more control points were determined taking as reference, the orthophoto maps or satellite images QGIS. This methodology was adapted from Crespo Sans, Dávila Martínez, and Camacho Arranz for the georeferencing of old maps [32–34].

The series of georeferenced maps were then used as a base to distill vector geometries related with water management structures (Figure 9).

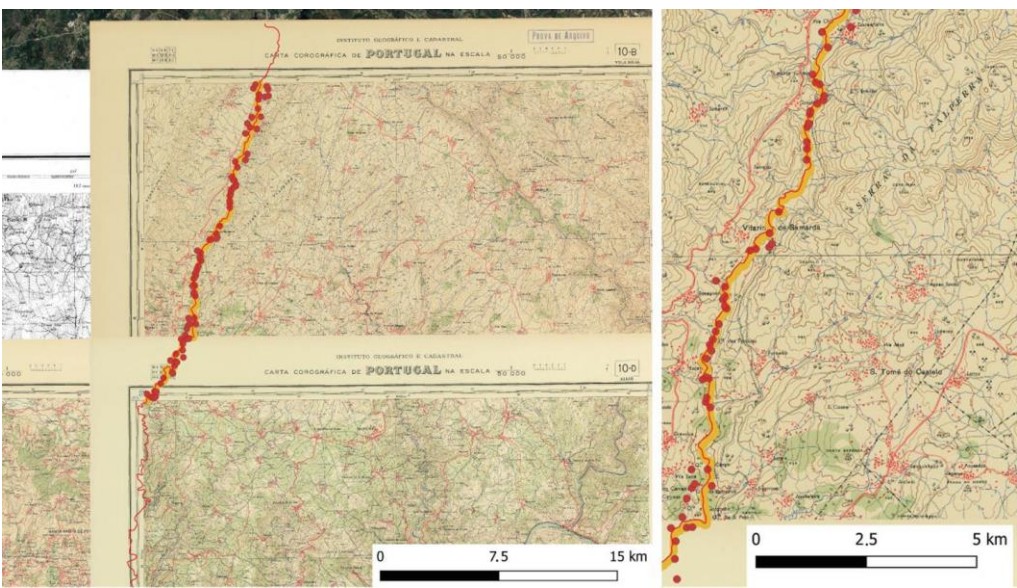

**Figure 9.** Georeferenced military old maps of Portugal and digitalization (vectorization) of the water cultural heritage.

Although the older cartography is not so fruitful in signalling elements of anthropic character related to the "Water" issue, some discrepancy was noted regarding the tracing of rivers, streams, and small watercourses, most probably due to technical difficulties (not-very-accurate topographic equipment and/or poor surveying conditions). Despite these discrepancies, already in the cartography of the 1930s, the presence of different types of water-related features is notorious.

## 5. Photogrammetry, Cartography, and Webgis (Google Earth and Google Earth Studio)

In essence, the present work seeks to integrate several approaches to new information technologies and the so-called digital humanities. The digital data management component, the digitization of heritage elements present in cartography, and, as already mentioned, the digitization of the territory with the use of online tools and ROV/drone equipment are important technical and methodological aspects of the study.

The integration, even if partial, of these technologies and the data obtained through them in a digital model, not abstract, of the territory and its human occupation brings an alternative to the visualization and understanding of the space in its medium scale, in a regional and local scale. The Google Earth Studio and QGIS tools, used to obtain georeferenced satellite images, allow us, through the transformation of these georeferenced images into digital terrain models, to perform the manipulation, visualization, and lighting and texture changes of these digital models through the open source software Blender.

The satellite images, aerial photographs or DEM, obtained through the Google Earth Studio platform, and the various drone flights and respective photographic collection allow us, overall, to obtain a complementary digital terrain model of about 2035 km$^2$. This comprehensive digital model was obtained through the Agisoft Metashape software, where each of the 101 images from the Google Earth Studio mosaic was analyzed and

autocorrelated, giving, as a result, a georeferenced digital terrain model, which it was possible to integrate, later, in QGIS after a DEM was obtained. This DEM was then exported with the extension "*.obj" to a modeling and digital treatment software, in this specific case, Blender. It was on this platform or application that the general and realistic image of the space, the territory, the orography, and the positioning of the vectorized anthropic data, from military cartography and historical cartography, took shape (Figure 10).

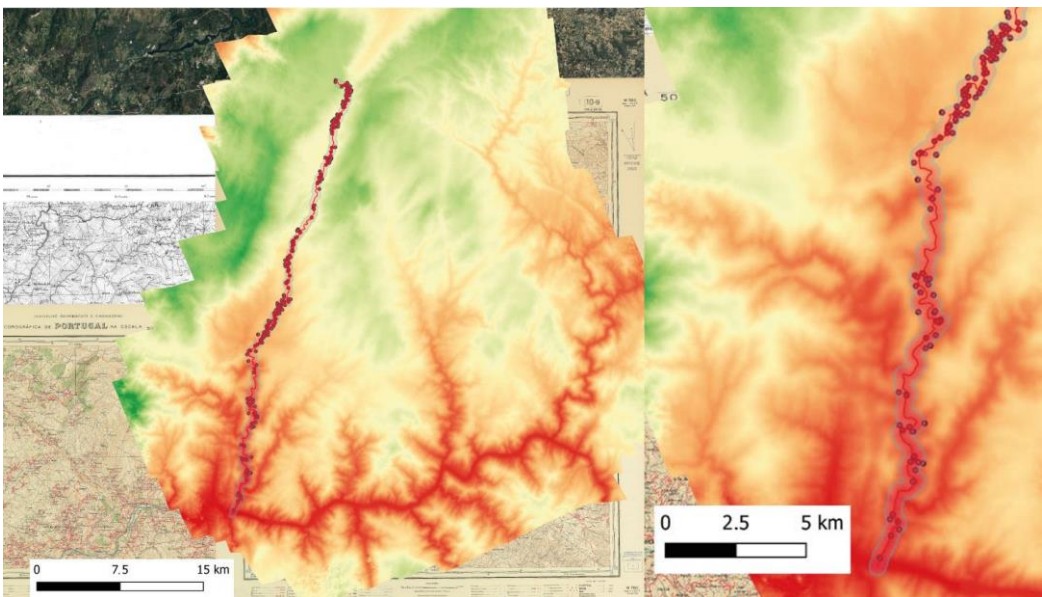

**Figure 10.** Overlay in QGIS of the old cartography, the mosaic of Google Maps images, and the digital terrain model obtained through Agisoft Metashape.

Although the images obtained through the Google Earth Studio and Google Earth web platforms do not possess specific characteristics for chromatic analysis due to the fact that these same mosaics are the junction of several capture sources (aerial images and photographs captured through flights, satellite photographs, and digital terrain models), the images captured by drone (PHANTOM 3 4K and Mavic Air 2) of parts of the course of the Corgo River did, however, allow some image processing to be carried out to highlight various aspects that facilitate the identification and interpretation of elements of anthropic character.

## 6. Results

The result of this work can be framed in four fundamental points: (1) obtaining two digital models of the territory or area under study using, essentially, two diversified resources: on the one hand, the use of images or mosaics of images from Google Earth Studio and Google Earth platforms, and, on the other hand, the use of aerial photographs obtained through ROVs or drones to obtain a mixed model, of good definition, of the territory under study; (2) creating a digital model of the physical space that could be treated in digital modelling applications and software, with the possibility of implementing digital tools of illumination, relief increment, textures, and other functionalities in order to allow a not-so-abstract understanding of the territory under study (Figure 11); (3) using geographic information systems applications or software—in this specific case, QGIS—to georeference old military cartography for the studied area and, from this process, to vector or digitize points of interest—in this specific case, built heritage—directly or indirectly related to the use and management of water resources; and (4) designing a mixed digital model of the territory and the orographic landscape (Figure 11), carried out, in essence, with open-source resources and its integration into a virtual model which, naturally, serves as a basis for

further studies. The georeferenced digital model's integration and transformation into DEM can, however, be used in the project prepared in QGIS (Figure 10).

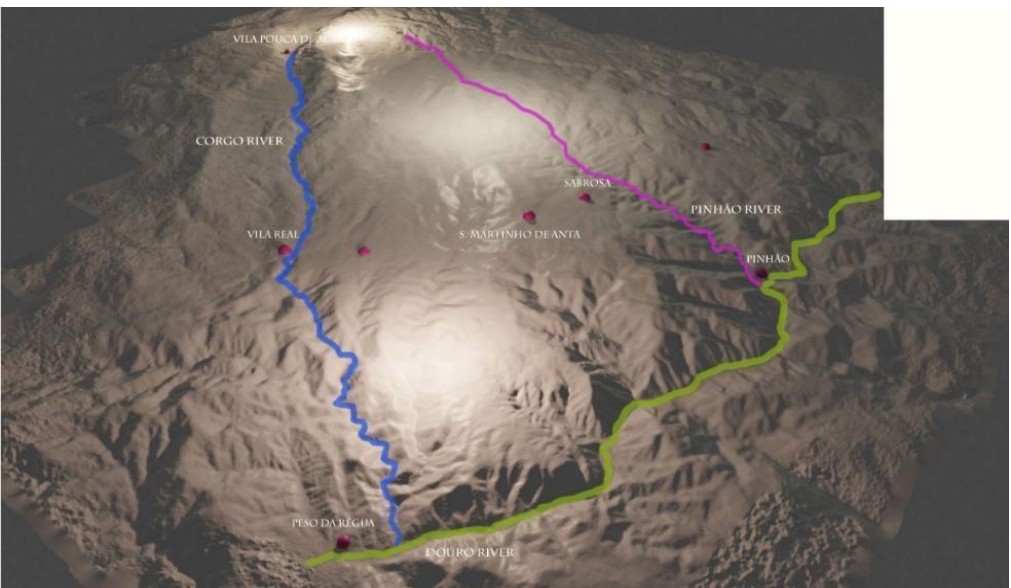

**Figure 11.** Digital model of the general area under study with texture and lighting treatment, obtained using the Blender software.

We call water-related heritage or water heritage to that heritage associated with water resources—catchment, storage, or management-, but also that which made it possible throughout history to mitigate the limitations of paths, routes, trails, and communication routes imposed by rivers, lakes, seas, etc. The bridges and water containment structures are fundamental for understanding the use and changes in the landscape and territory and their management.

The digital humanities and the domain, even if not very deep, of the available resources and tools, especially open-source resources, allow us, if used well and with good practice guides [11,14,35–37], to obtain a quite significant perception of the area under study and, evidently, as already mentioned, a less abstract structuring of the reality.

The mere notion of the evolution of the data in the historical documentation, in this specific case, of a part of the historical cartography, also allows us to reflect on the importance given to the elements of the physical and anthropic world referenced in that same documentation. In the work carried out, especially in the analysis of the references to the heritage related to the water on the banks of the Corgo River, we noted, as illustrated in Figures 9 and 12 and Table 1, an evolution in the number of elements referenced in the documentation and digitized/vectorized. This does not mean that the increase in the number of watermills, for example, is due to 29 watermills built from 1934 to 1963, but that most of them were not mapped in 1934. This is another aspect mentioned by some authors who naturally deal with the problems of using old cartography for studies on cultural and natural heritage [35,38].

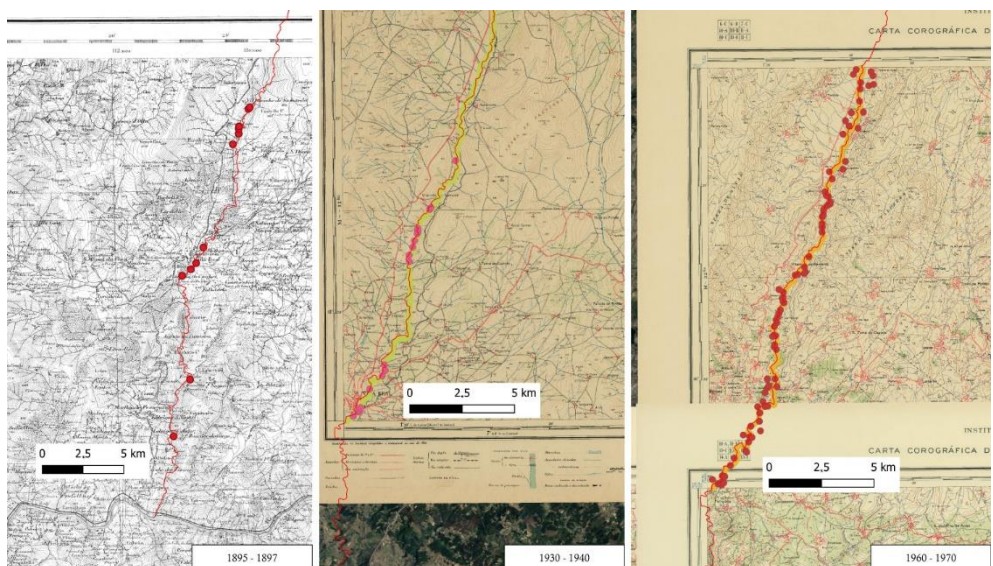

**Figure 12.** Sequence of historical military cartography with vectorization of heritage elements.

**Table 1.** Elements represented and identified in the historical cartography according to the years 1895, 1934–1935, and 1963.

| Year | Wells | Watermill | Fountains and Tanks | Spring/Water Mines | Norias | Bridges | Hydroelectric Power ST. | Total |
|------|-------|-----------|---------------------|--------------------|--------|---------|-------------------------|-------|
| 1963 | 53 | 10 | 11 | 0 | 7 | 7 | 1 | 99 |
| 1934 | 20 | 0 | 0 | 0 | 0 | 14 | 0 | 34 |
| 1895 | 12 | 0 | 0 | 0 | 0 | 0 | 0 | 12 |

On the other hand, the relevance of these processes and the various free platforms for the management, vectorization, and obtaining of digital models, both of the landscape and of the hydraulic built heritage, for education and heritage education naturally stands out. It is on the relevance of the hydric cultural heritage, the concepts of heritage, and its evolution that some authors highlight the various current potentialities [39,40]. The same authors also highlight the values and weight of heritage, and its integration into the landscape and the environment for education and, essentially, for heritage education [39]. In fact, nowadays, it is essential to complement theoretical resources with new technological realities; the so-called digital humanities are, necessarily, a fundamental resource and a field in constant evolution.

## 7. Discussion

"Water Heritage", "Water Landscape", and "Water Architecture" are all fundamental terms for understanding and caring for this human heritage (Figure 13).

In this framework, our research has sought to open up some dynamic perspectives on the use of new technologies and augmented reality in the fields of digital humanities, mainly open access, to show more realistic data on the area and the elements under study, but also simulations that help understand the landscape in different conditions of the river's flow. Digital abstraction and simulation amplify our own understanding of a given reality, especially when we analyze the landscape on a local and regional scale.

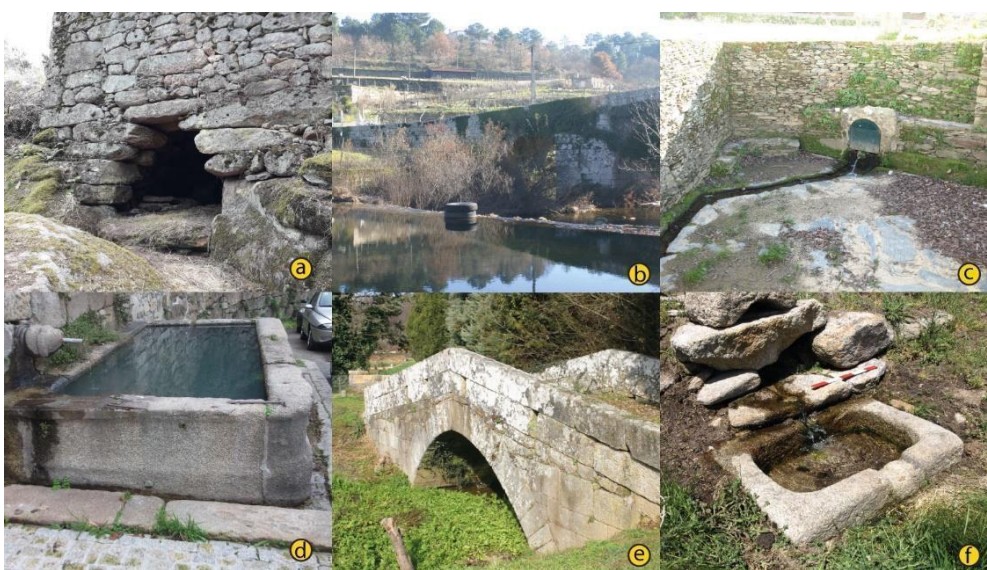

**Figure 13.** (**a**) Moinho de Anta; (**b**) ponte de Piscais; (**c**) fonte de Santa Marinha; (**d**) fonte da rua Marechal Teixeira, em Vila Real; (**e**) ponte medieval de Torneiros; (**f**) fonte da Senhora da Fraga.

The multiplicity of factors, natural, orographic, topographic, and anthropic, among others, limit, in a general way, our capacity to understand and interact with the environment that surrounds us. Our small relative scale is one of the factors that prevents us, in part, from obtaining an overview of the world around us. This, combined with the past issues, the territory's occupations, and the practical needs of ancient communities, makes our understanding even more difficult.

Simulations on the water level in certain periods of history help us to know about the configuration in the territory of the buildings and constructions and the distance to water and to better understand the effects and impacts of flood. It also allows an understanding of the movements of populations along riverbanks.

Mixed realities, new digital tools, and the modeling of space in a powerful and rigorous way facilitate our understanding of the management and handling of the environment, the landscape, and the territory over time.

Used in the archeology of landscape and archaeoastronomy, especially in studies on petroglyphs [41], we have seen how useful these technologies and applications can also be for studying and comprehending the water heritage and landscape. They help us to blend in a "mindscape" the archaeological landscape (what we can observe now of the past that is complex and decontextualized), with the ancient landscape (what we can imagine of the dynamic relationships of all its elements).

Mixed realities, new digital tools, and the modeling of space in a powerful and rigorous way facilitate our understanding of the management and handling of the environment, the landscape, and the territory over time.

In this sense, the mixed reality was implemented in our research to show and interpret the water heritage and landscape and to analyze and answer important historical and archaeological hypotheses about it.

## 8. Conclusions

The new information technologies and digital humanities, even with more than 60 years of existence [13,14,41,42], continue to offer new approaches to data treatment and interpretation. The possibility of representing, in a more realistic and assimilable way, the areas and objects of study is, undoubtedly, a fundamental resource.

In this work, the importance and relevance of new tools for capturing, managing, handling, and modeling digital data on heritage water and water landscape was highlighted.

The understanding of the alterations or changes that occurred throughout the history of humankind, especially in more recent periods, between the Early Middle Ages, the Late Middle Ages, the Modern Period, the Proto-Industrial Period, and, more significantly, already in the industrial revolutions, requires not only the comparative analysis of written documentation, the criticism of sources, and the crossing of diverse data, but also the fundamental use of archaeology and the archaeological method [43].

However, archaeology today has a wide range of resources in the fields of digital humanities that enable a diverse set analysis and new approaches. Geographic information systems are fundamental for the vectorization of cultural and archaeological elements, their analysis in geographic databases, and their visualization in a small or medium scale. On the other hand, the techniques of photogrammetry and digital modeling of historical and archaeological structures enable their understanding and morphological and functional interpretation.

The integration of digital elevation models, landscape models, visualization, and management of vectorized geographic information, in this case, on the water heritage and on the visualization, in medium scale, of the natural accidents related to the hydrographic network are quite relevant and vital. Digital changes of natural factors such as the level of riverbeds and increases in flows can, as illustrated in Figures 6 and 7, complement our view on the changes caused in the territory and on the positioning of cultural, historical, and archaeological elements in the same territory.

The region of Trás-os-Montes and Alto Douro, framed, almost in its entirety, in the so-called Demarcated Douro Region and the Alto Douro Wine Region, classified by UNESCO in 2001 as a World Cultural Heritage site, is a region rich in natural and anthropic heritage, a region little studied from the historical point of view [44–46] and whose orographic, topographic, and cultural constraints assume a high relevance from the point of view of historical, archaeological, and environmental research.

Besides the vectorized information and modeling data of the territory and the landscape with the possibility of changing/modeling, in post-processing, the same digital models, it is also important to obtain databases integrated in geographic information systems and all the possibilities of a direct and indirect analysis (geographical and graphical) of the obtained data and their comparison through time.

Through this research on heritage and landscape, what is important to highlight is the relevance of digital humanities, open-source systems and platforms, and the use of new technologies for multidimensional data capture that produces augmented realities for widening our understanding.

**Author Contributions:** Conceptualization, G.V.G. and D.B.P.; methodology, G.V.G. and D.B.P.; investigation, G.V.G., D.B.P. and M.G.-U.; writing—original draft preparation, G.V.G., D.B.P., A.d.S.M. and M.G.-U.; writing—review and editing, G.V.G., D.B.P., A.d.S.M. and M.G.-U.; visualization, G.V.G., D.B.P., and M.G.-U. All authors have read and agreed to the published version of the manuscript.

**Funding:** This research was co-funded by (1) the Cámara Municipal de Sabrosa (2) FEDER and Junta de Extremadura (ref. GR10187 and IB20182), (3) by the programme "Ayuda del Programa de Recualificación del Sistema Universitario Español", modality "Ayudas para la recualificación del profesorado universitario funcionario o contratado", reference RP-08, (4) Learning Villages #C, CERV-2022-CITIZENS-TOWN-NT - Networks of Towns, Project n.° 101091134-LVIN-C and (5) Learning Villages: Citizenship, Entrepreneurship, Heritage & Environmental Education for Rural Sustainable Development. Ref. 2020-1-ES01-KA227-ADU-096064.

**Institutional Review Board Statement:** Not applicable for studies not involving humans or animals.

**Informed Consent Statement:** Not applicable.

**Data Availability Statement:** Not applicable.

**Conflicts of Interest:** The authors declare no conflict of interest.

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
