# Peer review of "Water Heritage in the Rural Hinterland Landscapes of the UNESCO Alto Douro Wine Region, Portugal: A Digital Humanities Approach"

_heritage, doi:10.3390/heritage6040196_

Round 1

Reviewer 1 Report (Previous Reviewer 3)

Fine/minor spell check required. Read the reviewed pdf.

Author Response

Response 1:

In the bibliography cited there are, however, several recent, if wide-ranging and methodologically diverse, articles on applied experiences and methodologies. Among these articles we highlight the cited article no. [10], no. [13], no. [9], no. [20], no. [27]. Although it is a pilot work, which uses, above all, open-source tools perfectly described and explained, in reality this type of approaches, using open-source platforms like google earth studio to obtain maps/images/mosaics to produce digital models through photogrammetry is not particularly cited in recent works. However, we will take into account these important reflections in further work.

Reviewer 2 Report (New Reviewer)

The manuscript reports findings of an interesting study. The research question is clear and the described methodology appears sound. There is one key aspect which the authors should work on when preparing a final version of the manuscript:

The (important!) discussion section lacks a connection to the state-of-the-art literature. There is no single reference to previous studies / publications. In the current version of the manuscript, there is no ‘take home’ message for the readership. In how far does your study (or at least your methodology) extend, back up or even contradict previous findings?

Author Response

Response to Reviewer 1 Comments

Point 1: The manuscript reports findings of an interesting study. The research question is clear and the described methodology appears sound. There is one key aspect which the authors should work on when preparing a final version of the manuscript: The (important!) discussion section lacks a connection to the state-of-the-art literature. There is no single reference to previous studies / publications. In the current version of the manuscript, there is no ‘take home’ message for the readership. In how far does your study (or at least your methodology) extend, back up or even contradict previous findings?

Response 1: In the bibliography cited there are, however, several recent, if wide-ranging and methodologically diverse, articles on applied experiences and methodologies. Among these articles we highlight the cited article no. [10], no. [13], no. [9], no. [20], no. [27]. Although it is a pilot work, which uses, above all, open-source tools perfectly described and explained, in reality this type of approaches, using open-source platforms like google earth studio to obtain maps/images/mosaics to produce digital models through photogrammetry is not particularly cited in recent works. However, we will take into account these important reflections in further work.

Round 2

Reviewer 2 Report (New Reviewer)

The authors provided a resubmitted version of their manuscript. They considered all points mentioned in review round no. 1. Their explanations are logically coherent. Against this background, I would like to recommend the current version of the manuscript for publication.

Author Response

Response to Reviewer # 1 Comments

Point #1: The authors provided a resubmitted version of their manuscript. They considered all points mentioned in review round no. 1. Their explanations are logically coherent. Against this background, I would like to recommend the current version of the manuscript for publication.

I think everything is OK. now with the article. Thanks for all the collaboration and attention. Thank you very much for your kind attention and dedication.

This manuscript is a resubmission of an earlier submission. The following is a list of the peer review reports and author responses from that submission.

Round 1

Reviewer 1 Report

The manuscript mentions the relationship between human culture, sites and water. However, it focuses only on the creation of DEM and alignment of old maps within the same GIS system. This is misleading. I expect to see at least a preliminary description on how you plan to describe and integrate sites, by their multiple nature - wells, mills, dams, irrigation systems, etc. 

you mention as well the use of extended reality, but there is no discussion on it. 

Reviewer 2 Report

The paper is very ambitious as it deals with various interconnected topical issues: digital humanities, the historical landscape (in particular the water and rural landscape of the Alto Douro region), 3D and GIS applications.

Each of these topics is very complex and requires a more exhaustive framework than that provided by the authors, both as regards the Introduction (which only outlines the "state of the art") and the bibliographic references, which at the moment appear too limited and meager compared to the numerous recent contributions.

The research path addressed by the authors is clear but too limited, or in any case illustrated too superficially precisely with reference to the aforementioned complexities. An in-depth study is not only desirable but necessary.

Even the results, although partial, are clear but limited. The authors have developed a tool that can be used by other researchers in the future but have not sufficiently clarified the limits or criticalities of the tools used (especially open source) or the possible possibilities of implementing other archival sources (historical iconography different from that military, historical registers and so on).

Reference 6 (line 349) is incomplete.

Between "figure 7)" and "3)" (line 257) a semicolon is desirable.

For all these reasons, a major review is proposed. However, I suggest to the authors to withdraw the paper and submit it soon in a more organic and complete version.

Reviewer 3 Report

The article is interesting overall. the methodology expressed has vast potential for the digital humanities in the near future. I especially find the use of images from Google Earth Studio interesting. For this reason, it should indeed be published.
Moreover, extensive editing of the English language and style is required. The "Lecturer style text" is not clear to the readers. Follow the suggestions given and include in that revision phase a native En speaker.